# Effect of Composition on the Physicochemical Properties of Cross-Linked Poly(sodium acrylate)/Sodium Silicate Hydrogels

**DOI:** 10.3390/ma16237422

**Published:** 2023-11-29

**Authors:** Joanna Mastalska-Popławska, Łukasz Wójcik, Piotr Izak, Damian Konaszewski

**Affiliations:** Faculty of Materials Science and Ceramics, Department of Ceramics and Refractories, AGH University of Science and Technology, 30-059 Krakow, Poland; lukwoj@agh.edu.pl (Ł.W.); izak@agh.edu.pl (P.I.); konaszew@student.agh.edu.pl (D.K.)

**Keywords:** hybrid hydrogel, fire retardancy, intumescent system, rheology, thermal analysis

## Abstract

The paper presents the results of research on the influence of the components of composition on the physicochemical properties of cross-linked poly(sodium acrylate)/sodium silicate hydrogels. The results of the rheological measurements showed that an increasing share of polyacrylate in the sample causes a shift of the cross-over point towards longer times and improves mechanical properties. In turn, increasing the share of sodium silicate has a positive effect on extending the fire insulation time (I). TG/DSC analysis indicated that if the sample contained more than 50 wt.% of polymer, the thermal decomposition was a three-stage process, while if the sample contained more water glass, the decomposition was two stage. UV-Vis measurements and SEM analysis confirmed, inter alia, that increasing the share of polyacrylate causes improvement of the optical parameters and homogeneity of the tested hydrogels. The presented results suggest that by controlling the share of the main components of the silicate–polymer hydrogels, it is possible to control their mechanical, optical and thermal properties. Multiplication of the layers or increasing their thickness improves their fire-retardant properties.

## 1. Introduction

The development of research on the use of soluble silicate solutions as binders for fire-resistant glass was initiated in the mid-20th century in the United States. Back then, only the sodium water glass solution was used, which, after being poured onto a glass pane, was dried to evaporate excess water (temperature around 100 °C) and then a second pane or a sheet of stiff transparent plastic such as polyvinyl chloride or polymethacrylate was glued. Talc, powdered glass fibers and silica were used as additives to thicken and improve the fire resistance of such glass. The binder connecting the individual glass elements was a water glass solution or epoxy resin [1]. Thus, created glazing could withstand temperatures of approximately 1000 °C. During contact with fire, water evaporated from the gel and the remaining salts crystallized, significantly increasing their volume and creating an opaque pumice, which at the same time, provided heat and fire insulation and was a binder connecting the glass panes. At the same time, a similar solution was proposed, except that in addition to sodium water glass, the following sodium or potassium salts were also added: aluminates, leadens, stannates, sulfates, aluminum sulfates, borates and phosphates. These salts, in combination with a sodium silicate solution, are able, after evaporation of excess water, to create transparent films [2,3] that show better fire resistance than the gel formed only from a sodium silicate solution.

Schaar et al. [4] proposed a gel with insulating properties that can also be used as a protective coating for wooden elements, synthetic carpets, plastic bottles, etc. The advantage of this composition is the small amount of smoke emitted when exposed to high temperatures. Admixtures of the following substances, fulfilling specific tasks, are added to the sodium silicate solution: ammonium phosphate—an agent that improves heat and fire resistance; urea and/or cyanoguanidine—under the influence of temperature, these substances decompose into gases, causing the swelling of the gel; sucrose—together with phosphates, initiates the swelling process at low temperatures; titanium dioxide—a substance ensuring low absorption of thermal radiation. Other solutions used in the 1970s were also based on the use of phosphates, chlorides, sulfates, etc., of metals such as zinc, magnesium and calcium. Sometimes they were additionally acidified with phosphoric acid to accelerate the gelation reaction [5,6].

In the early 1980s, De Boel proposed a new intumescent material for fire-resistant glass [7]. It was based on the works of Gaeth [1] and Schaar [4], but additional substances that improved the fire-resistant properties of the glass have been introduced. Additionally, he tested the effects of polyhydroxyl alcohols (glycerin, ethylene glycol, sorbitol), monosaccharides (glucose), and polysaccharides (starch), which improved the performance of fire-resistant layers based on sodium water glass solutions as follows:lowering the temperature at which fire-resistant pumice begins to form—sodium phosphate;extending the time during which the protective layer is modified at a constant temperature—glycerin, ethylene glycol, starch;reduction of the gradient of the time/temperature curve after which the swelling of the fire protection layer ends—glucose, colloidal silica, borax, boric acid, sodium phosphate, sodium aluminate, aluminum phosphate;extending the time interval during which the temperature remains essentially constant as gel swelling progresses—urea, glycerin, sorbitol [8,9,10].

Research conducted by De Boel showed that it is enough to add only a few percent (even less than 5%) of the above-mentioned substances to improve the properties of the fire-retardant gel.

In situ polymerization between glass panes was first used in 1989. In addition to the standard additives, anti-corrosion substances (alkaline phosphates, tungstates, molybdates) were also introduced into the acrylamide-based polymerization mixture [11]. Then it was poured between panes covered with metal tape. After about 30 min, a transparent hydrogel was obtained. However, as we know, as a result of factors such as temperature or UV radiation, polymers are subject to degradation, i.e., aging, that is manifested, among other things, by modifications to their surface, i.e., matting, cracking, color change. In 2015, Litovchenko et al. [12] published an article describing methods of modifying pure acrylic hydrogel constituting a fire-retardant layer in a glass pane that was cross-linked with hexamethoxymethyl melamine ether. The elasticity of the gel was controlled using this additive. The more flexible and less fluid it was, the greater the fire resistance of the glass pane was. Zhang et al. [13,14] investigated hydrogel polyacrylamide-based fire-retardant layers with the addition of salts such as magnesium chloride and potassium aluminum sulfate, which had a similar effect to the hexamethoxymethyl melamine ether. Liu et al. [15,16] tested water glass systems with a molar modulus of 4.0 with the addition of glycerol and silica, where the gelling agent was ammonia, and the resulting glass packet was heated at 80 °C for 1 h to gel. A similar polymerization method was described already in 2000 by Sugiura K. et al. [17]. Polymerization took place in a sodium water glass solution, where 30 wt.% was silicon dioxide. Due to the high alkalinity of the medium, instead of acrylic acid, which is very sensitive to pH changes, acrylamide derivatives, which form a transparent gel under such polymerization conditions, were used as a monomer [18]. N,N′-methylenebisacrylamide and its derivatives were used as the cross-linking monomers. Additionally, several percent admixtures of metal oxides (CaO, MgO, PbO) were introduced into the mixture in order to improve the insulating properties, as well as photoinitiators (derivatives of acetophenone, acetals, pyrene, etc.) and photosensitizers (aromatic and aliphatic amines) of the polymerization reaction because the gel was hardened with UV radiation. An anti-freezing agent was also used to prevent the gel from freezing at low temperatures—ethylene glycol, glycerin, etc. A similar solution, where a mixture of silanols and sodium silicates was subjected to a polymerization reaction, is presented in the Birkhahn’s [19] and Gelderie et al. [20] patents.

Work on new types of fire-resistant glass panes carried out after 2000 mainly covers two research paths—polymer–silica hydrogels with a medium in the form of soluble silicate solutions and gels based on organic polymers. Additionally, research on the influence of aqueous colloidal solutions as a factor increasing viscosity through the possibility of creating spatial structures using polymer chains is being carried out. This is due to the presence of various functional groups that can interact with other molecules or fragments of polymer chains. Mixtures of polysaccharides, polyhydric alcohols, etc., are used as fillers. After introducing the aqueous colloidal solution into the mixture containing soluble silicates and other components of the polymerization feed, such as monomer (here—derivatives of acrylamide and acrylic acid) and polymerization reaction initiators, it is possible to harden the sol with thermal polymerization (cast in place) [21,22].

Based on our previous works [23,24,25], silicate–polymer hydrogels, where the silicate part was sodium water glass and the polymer part was sodium polyacrylate, were used as the research material. As a polymer component, from among the available vinyl monomers such as acrylic monomers, methacrylates, vinyl acetates and vinyl alcohol, we chose sodium acrylate due to the fact that its polymer can bind significant amounts of water and polymerize in the presence of other chemical substances, and the polymerization process itself is not complicated. In the case of the silicate component, we chose sodium water glass because it is non-toxic, cheap and easily available, and during cross-linking of vinyl polymers, silicon gel precipitates from it and a mineral–organic hydrogel with increased durability and elasticity is created [26]. A mixture of sodium polyacrylates of various molecular weights was used as a potential additive increasing elasticity. Transparent hydrogels with increased mechanical properties were obtained. The received test results made it possible to determine the impact of individual components of the tested hydrogels on their physicochemical properties, including fire-retardant properties. Another advantage of the developed hydrogel is the fact that the system of redox initiators we use allows for lowering the polymerization temperature, which is useful when filling fire-resistant glass, and additionally reduces the costs of the process. The presented test results will contribute to the development of the science in the field of fire-resistant materials based on silicate–polymer hydrogels.

## 2. Materials and Methods

### 2.1. Materials

The following reagents were used to prepare the silicate–polymer hydrogels:source of sodium silicate species—sodium water glass R-150 (WG) with a molar modulus of M = 2.05 and a density of 1.51 g/cm^3^ (Rudniki Chemical Plant, Rudniki, Poland);vinyl monomer—aqueous solution of sodium acrylate (ANa) with a percentage concentration of 15 wt.% and 20 wt.%, synthesized in the laboratory. It was prepared using acrylic acid (stabilized, 99.5%, density of 1.05 g/cm^3^ (Acros Organics, Geel, Belgium)) and sodium hydroxide microgranules (Avantor Performance Materials, Gliwice, Poland);polymer additive—a mixture of sodium polyacrylates(R-100) of various molecular weights with a density of 1.29 g/cm^3^ called Midafen R-100 (Lubrina S.A., Łódź, Poland);redox initiators of polymerization reaction—potassium persulfate (KPS; Acros Organics, Geel, Belgium) and sodium thiosulfate (NTS; Acros Organics, Geel, Belgium);cross-linking monomer—N,N′-methylenebisacrylamide (NNMBA; Acros Organics, Geel, Belgium)All reagents were used without further purification.

### 2.2. Samples Preparation Procedure

The sodium acrylate solution was prepared as described in [19] and diluted to the proper value, i.e., 15 wt.% or 20 wt.%.

In the first stage of silicate–polymer hydrogels preparation, sodium water glass was mixed with the polymer additive for about 10 min using a magnetic stirrer. Then, sodium acrylate solution was gradually added to the prepared mixture and mixed again until a transparent solution was obtained. The next step was to introduce a cross-linking monomer (NNMBA) into the mixture. The last stage of hydrogel preparation was the addition of redox initiators (NTS and KPS). After introducing both NNMBA and the NTS/KPS system, the mixture was stirred each time until the mentioned substances were dissolved. The composition of individual samples is summarized in Table 1.

### 2.3. Characterization Methods

The gelation kinetics study and tests of viscoelastic properties of the silicate–polymer hydrogels were performed using an Anton Paar Physica MCR-301 rheometer. The kinetics measurements were carried out in the PP25 system (plate–plate with a diameter of 25 mm), with a constant gap height of 0.2 mm and at ambient temperature. The angular deflection was 0.5% and the frequency was 1 Hz. The study was conducted immediately after adding the initiators to the system for 40 min. The results were presented on graphs of the dependence of the storage modulus (G′) and the loss modulus (G″) on time in a logarithmic system. Tests of the oscillation properties were also carried out in the PP-25 system, on the gelled samples in the shape of thin discs with a diameter of about 3 cm and a thickness of 2 mm, prepared 24 h before the test (samples were cast in the round plastic molds with a diameter of 30 mm). The height of the measurement gap was set at 2 mm, and the entire test was conducted for 10 min at ambient temperature, with a constant amplitude of 0.3% and a frequency of 1 Hz. During measurement, it was observed whether the values of the G′ and G″ moduli were changing. Based on the average value of the moduli after establishing a plateau, the value of the phase shift angle was calculated according to the following formula:(1)$$\tan\varphi =\frac{G"}{G'}$$
where: -G′—storage modulus [Pa],-G″—loss modulus [Pa],-tan*φ*—tangent of the phase shift angle [−].

Rheological measurements were performed three times for each sample. The presented results are the average value.

DSC/TG thermal analysis tests were carried out using the NETZSCH STA 449 F3 thermal analyzer (NETZSCH, Selb, Germany) Measurements were carried out in an air atmosphere, in aluminum crucibles, at a heating rate of 10 °C/min. Endothermic and exothermic effects were determined based on the DSC curves. The thermogravimetric curve was used to determine the mass loss of the sample.

The fire tests were performed with the use of a research stand for fire tests which consists of a furnace equipped with an infrared camera (NEC Thermo Gear G100 (NEC, Tokyo, Japan) and a removable torch fueled with propane-butane. Additional elements are two thermocouples (inside and outside the furnace) and an exhaust system. Samples of the dimension of 20 cm × 30 cm were examined for the average temperature using a burner panel. The camera was placed 2 m from the test stand. Hydrogel-filled glass samples were prepared as follows:glass forms were prepared from two glass panes with dimensions of 20 cm × 30 cm × 0.4 cm connected with a 1 mm thick silicone seal. The use of a silicone gasket allowed for the creation of a gap between the glass panes, in which the fire-resistant hydrogel was placed. The edges of the prepared forms were additionally protected with a high-temperature silicone to prevent the gel from flowing out during the test;immediately after the preparation of the hydrogel, it was introduced into the prepared forms, which were additionally sealed with aluminum tape at the infusion point (Figure 1);the samples prepared in this way were seasoned at room temperature for 7 days before testing.

As a result of the measurements, fire insulation (I), was determined [27]. Fire resistance (E) and fire load capacity (R) were maintained in each trial, so they were not analyzed. 

SEM/EDS analysis was intended to describe the structure of the analyzed samples of the silicate–polymer hydrogels, taking into account their qualitative composition, and to determine porosity. The NOVA NANO SEM 200 electron microscope (FEI Company, Hillsboro, USA) with an EDS analyzer was used for the research. The test was conducted in a low vacuum, without sputtering the samples. Hydrogel samples were tested before and after the fire test. Hydrogel samples before the fire tests were dried at room temperature, frozen using liquid nitrogen and immediately crushed in a metal mortar. They were then dried again at room temperature and tested. Pumice, which was used for samples after the fire tests, was collected from between the glass panes and then tested.

The translucency tests of the silicate–polymer hydrogels were carried out using Philips PU-8750 UV-Vis (Philips, Eindhoven, Netherlands) spectrophotometer, in the range of 200–1100 nm. Hydrogel samples were polymerized between 1 mm thick glass plates. The distance between them was 2 mm and 4 mm, respectively. Single-layer samples were tested. Plates without the gel filling with a distance of 2 mm were tested as a reference sample. Selected samples were aged for 24 h before testing using a diode UV lamp with a power of 0.3 mW/cm^2^.

## 3. Results and Discussion 

### 3.1. Rheological Measurements

#### 3.1.1. Gelation Kinetics

Figure 2 shows exemplary curves of changes in the storage modulus G′ and the loss modulus G″ depending on time for samples with the highest content of sodium water glass and 20 wt.% of aqueous solution of sodium acrylate in the composition (rest of the curves can be seen in the Appendix A), while Figure 3 shows the average values of the G′ and G″ moduli that were achieved during measurement and the designated cross-over time. For most samples with a mass ratio of water glass to sodium acrylate solution of 1:1 and 9:1, the G′ modulus value exceeded the G″ modulus value at the initial stage of measurement, which proves that these samples gelled immediately after adding the initiator. This behavior indicates that high pH, here coming from water glass, accelerates the gelation of vinyl systems, such as sodium acrylate [28,29,30,31]. The cross-over point indicating the beginning of the gelation process was determined for the remaining samples. This was the moment of intersection of the G′ and G′ moduli, and after that, the value of these moduli started to increase rapidly, and the values of the storage modulus began to exceed the values of the loss modulus. For samples 5%/1:9 and 20%/1:9, the gel point was reached in a shifted time range, i.e., 65 and 130 s respectively. It can be assumed that increasing the share of polymer additive causes the gelation point to shift to higher time values. For the samples with a 15 wt.% aqueous solution of sodium acrylate, a cross-over point was observed in each case (starting from the samples with the highest polymer content, this parameter was as follows: 80 s, 150 s, 210 s, 110 s, 315 s, and it was shifted in time compared to the corresponding samples with a 20 wt.% aqueous solution of sodium acrylate, which confirms the fact that the more diluted the solution of the starting monomer, the later polymerization occurs [32,33].

To sum up this part of the research, taking into account the possible future industrial use of the discussed hydrogel mixtures, it should be emphasized that the most favorable parameters characterizing the system would be an extended cross-linking time, allowing the filling of the fire-resistant glass mold without fear of premature gelation and sufficient deaeration.

#### 3.1.2. Oscillation Measurements 

Figure 4 shows exemplary graphs obtained during the oscillation test (rest of the graphs can be seen in the Appendix A), while Table 2 shows the average results obtained on the basis of the calculations described in Section 2.3.

Analyzing the graphs obtained during the oscillation measurement, it was noticed that a plateau was established in the initial 100 s of the measurement. Comparing the average values of the G′ and G″ moduli depending on the monomer solution concentration, values several times higher were found in the case of a more concentrated solution, which proves that sodium polyacrylate gives mechanical strength to the discussed hydrogels. However, no trend was found depending on the composition of the samples, i.e., comparing the share of total polymer and water glass content in terms of the phase shift angle values. The obtained hydrogels are not liquid, and due to the presence of a three-dimensional polyacrylate network, even in the minimum amount for the 5%/9:1 sample, they are highly flexible materials. If this mixture was stripped of the cross-linking agent, a highly viscous liquid would be obtained, with little evidence of gel structure from the precipitated silicate gel.

All of the tested systems achieved a phase shift angle value well below 45°, which means that the tested gels are elastic and the viscous component is small. If the samples obtained an angle value above 45°, it would mean that the liquid segments outweighed the sample composition [34]. 

### 3.2. Fire Tests

Table 3 includes fire insulation times (I), which were determined on the basis of the thermograms from a thermal imaging camera. In turn, Figure 5 and Figure 6 show, respectively, exemplary thermograms obtained during the fire test and graph of the dependence of temperature changes versus time used to determine the I parameter during the fire test (rest of the graphs can be seen in the Appendix A).

Analyzing the influence of the composition on the fire insulation results, it was noticed that the more water glass was in the sample, the longer the fire insulation time was. This can be explained by the fact that water glass is a fire retardant, which, by losing water and transforming into a swollen pumice upon the contact with high temperatures, extends the insulation time. However, the best results were obtained for the samples where the content of the water glass and sodium polyacrylate was similar, which allows the assumption that the structure of the resulting hybrid hydrogel may also have an impact on fire insulation, and in this case, it was the most durable (mechanical strength has been confirmed with rheological tests). The thickness of the gel layer may also have an impact on fire insulation time. The thicker it is, the longer this time will be [15,35,36].

It was noticed, when analyzing compositions with 15 wt.% and 20 wt.% of aqueous solution of sodium acrylate, that both gel systems increase significantly in volume (up to 5 times) during measurement (Figure 7). They start to soften and “boil” at a temperature of about 70–75 °C. The more water there was in the gel, the greater this effect was. However, the gel turned into foam very quickly, and at a temperature of about 90 °C, a solid began to form and fire-resistant pumice was created (Figure 8). At a temperature of about 80 °C, the so-called plateau was observed, which is confirmed in Figure 6. It can be concluded that in the case of the developed silicate–polymer hydrogels, the gel melting effect cannot be eliminated, but it is very limited.

### 3.3. TG/DSC Analysis

Figure 9 shows exemplary thermograms with TG (change in mass loss) and DSC (thermal effects occurring during heating) curves (rest of the thermograms can be seen in the Appendix A), while Table 4 summarizes the obtained results of mass loss and temperature related to the exothermic decomposition of the tested silicate–polymer hydrogels.

The greatest weight loss, over 50%, was observed for the samples in which the sodium acrylate solution content was 90%. This is related to the greater share of water in the hydrogel, coming from both sodium polyacrylate and sodium silicate solution, and its evaporation to approximately 200 °C (endothermic transformation barely visible on the DSC curve and inflection on the TG curve). TG/DSC analysis of these samples also shows that further thermal degradation is a two-stage process, and in the range of 430–500 °C two exothermic peaks appear, responsible for the degradation of the polymer, cross-linker and initiators system. For the remaining tested samples, mainly two-stage decomposition was observed (Figure 9b). The more sodium silicate there was in the sample, the smaller the mass loss. No glass transition of the polymer was observed for any of the analyzed samples (endothermic effect, for this type of silicate–polymer hydrogel occurring at a temperature of approximately 150 °C). This was caused by the overlap of thermal effects resulting from the transformation of both hydrogel components. Comparing the share of 20 wt.% and 15 wt.% of sodium acrylate solution, no major differences were found in either the weight loss or the decomposition temperature of the polymer itself [37,38,39].

### 3.4. UV-Vis Analysis

Three compositions of the silicate–polymer hydrogels were tested using a UV-Vis spectrophotometer:(a)composition 5%/1:9 with 20 wt.% aqueous solution of sodium acrylate,(b)composition 20%/1:1 with 15 wt.% aqueous solution of sodium acrylate,(c)composition 5%/1:9 with 15 wt.% aqueous solution of sodium acrylate.

The above-mentioned compositions were tested while changing two parameters, i.e., gel thickness and aging time:a—without aging, gel thickness 2 mm,b—aged for 1 h, gel thickness 2 mm,c—aged for 24 h, gel thickness 2 mm,e—no aging, gel thickness 4 mm,f—aged for 1 h, gel thickness 4 mm,g—aged for 24 h, gel thickness 4 mm.

No visible changes were noticed in the gel structure (turbidities etc.) after aging tests. However, in some samples, in the place of air access (upper part of the sample with the inscription), shrinkage and slight turbidity were noticed on the surface, most likely caused by water loss and chemical reactions occurring at the gel/air interface (Figure 10). 

The sample without hydrogel between the glass plates is marked as 0 in Figure 11, Figure 12 and Figure 13 which show the UV-Vis spectra.

Analyzing Figure 11a, it was found that most of the measured compositions are characterized by a better translucency than glass itself (85%). Only in the case of composition f (hydrogel of 4 mm thickness and aged for 1 h), translucency (approx. 80%) is worse in both cases, in comparison to clean glass and other samples (approx. 90%). In the case of 5%/1:9 with 15 wt.% aqueous solution of sodium acrylate hydrogel, compositions e, f and g (4 mm thickness) are characterized by a better translucency than pure glass and samples a, b and c (2 mm thickness) (Figure 11b), but to a lesser extent than in the case of a hydrogel with a higher sodium acrylate solution content. In the case of the 20%/1:1 with 15 wt.% aqueous solution of sodium acrylate hydrogel composition, all of the samples, except sample a (no aging, 2 mm gel thickness), are characterized by a better translucency than pure glass (Figure 11c). However, what is puzzling is that the sample which was unaged, is characterized by the worst parameters. This result may have been influenced by the fact that the sample could be, for example, filled with air. Summing up, there is little effect of UV radiation on the translucency of the analyzed polymer–silicate hydrogels. However, it is still marginal, which allows to assume that the analyzed hydrogels are resistant to UV radiation and presumably could last longer under conditions of normal use as a fireproof glass filling (where the glass thickness is at least 4 mm) for at least 5 years. But to be sure of this statement, more research needs to be carried out in this direction. 

Analyzing the ratio of water glass to sodium acrylate solution, it can be concluded that the best optical parameters have a composition with a higher acrylate content (5%/1:9, 20 wt.% of ANa aq.) than the relative composition, but with 15 wt.% of sodium acrylate solution (5%/1:9, 15 wt.% of ANa aq.), and the worst optical parameters are characterized by a composition with the same amount of water glass and sodium acrylate solution, i.e., 20%/1:1 with 15 wt.% of ANa aq.

### 3.5. SEM Analysis

Figure 12 show SEM micrographs of the selected samples of the tested hydrogels before fire testing (after drying).

Based on the photos of the hydrogel samples after drying, it was difficult to determine the relationship between the amount of water glass or polymer. It was only noticed that the more polymer there was in the sample, the smoother the sample surface was. However, when there was more water glass in the sample, the presence of lighter, oval-shaped fields was found (Figure 12c), which may be the precipitated silica. This may suggest that gelation of the tested systems occurs not only by polymerization, but also by the precipitation of the silicate forms which is influenced by the changing pH. However, no open porosity was found in the examined samples; therefore, SEM micrographs of the hydrogels after the fire test were also taken (Figure 13). Some forms of porosity can be seen in the photos, which may suggest that the tested hydrogels are characterized by closed porosity. During contact with fire, after the water evaporates, the resulting pumice undergoes transformation. Pores enlarge, merge, and then may be damaged under the influence of external factors, such as pressure. Due to the fact that sodium polyacrylate, which is the scaffold of the obtained hybrid hydrogel, has an amorphous structure, the resulting swollen layer has an irregular morphology [29,30,31,40,41].

Based on EDS analysis, it was found that the tested ash consisted only of silicate salts and carbon formed after combustion of the polymer (Figure 14).

## 4. Summary

Transparent silicate–polyacrylate hydrogels with increased fire resistance, which can be used both as a gel layer in fire-resistant glass and as an impregnation for other building materials, increasing their fire-resistant properties, were obtained. The results of the thermal analysis showed that during thermal degradation, water is released first from both main components of the hybrid mixture, i.e., sodium water glass and sodium polyacrylate (up to 200 °C). The polymer decomposes at temperatures above 400 °C, while silicate structures undergo only polymorphic transformations. Fire tests allowed for the assessment of compliance with the requirements specified in the testing standards. The tests showed that the best fire insulation is achieved by the samples with a mass ratio of water glass to a polyacrylate of 1:1 and higher. In these samples, the gel filling swelled more, which resulted in an increase in the specific surface area of the sample and, therefore, in an increased heat transfer. By multiplying such layers or increasing their thickness, it is possible to obtain a building material with the desired fire-resistant properties. Gelation kinetics studies have shown that an increase in the addition of sodium water glass causes faster gelation of the system (due to an increased pH), while if more polyacrylate is introduced or the starting monomer is more diluted, this will prolong the process. The increased polymer content also improves mechanical properties, i.e., elasticity (increase in the G′ modulus value), and also affects the homogeneity of the hydrogel (additionally confirmed on the basis of SEM analysis).

The presented research results are the basis for further application plans. In the next stage, it is proposed to carry out tests for the use of other water glasses, not only sodium, and various additives supporting both cross-linking and, for example, frost resistance or resistance to UV radiation.

## Figures and Tables

**Figure 1 materials-16-07422-f001:**
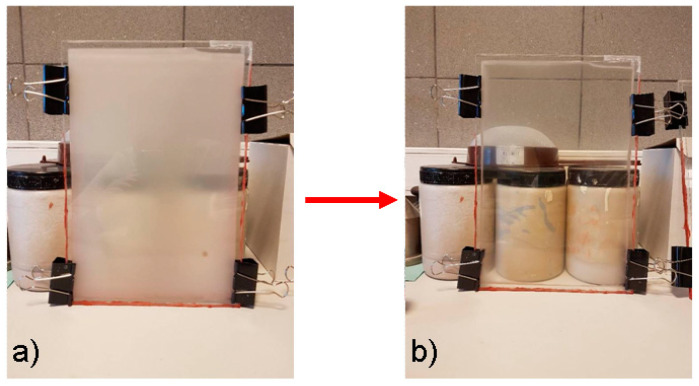
Glass mold immediately after filling with the hydrogel (**a**) and 24 h after preparation (**b**).

**Figure 2 materials-16-07422-f002:**
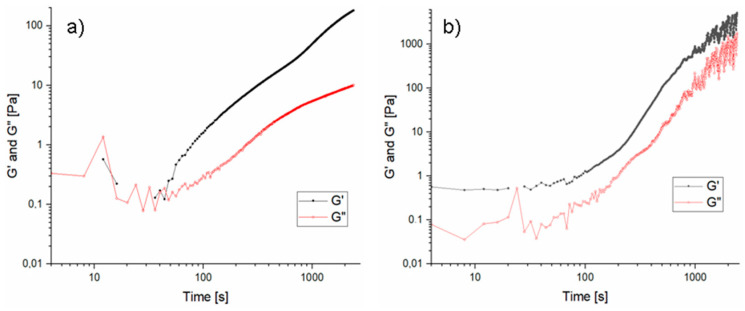
Exemplary graphs of gelation kinetics of silicate–polymer systems: (**a**) 5%/1:9 and (**b**) 5%/9:1 (20 wt.% of ANa aq.).

**Figure 3 materials-16-07422-f003:**
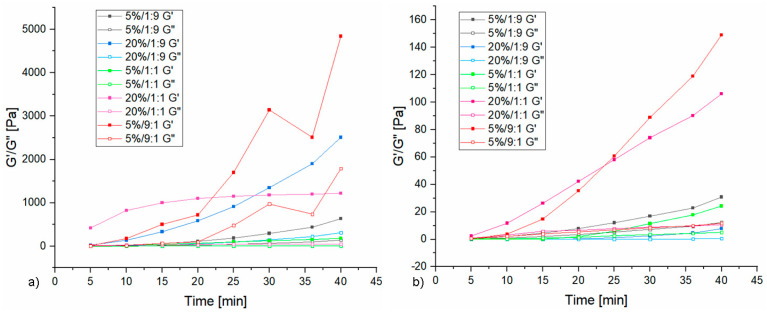
Dependence of the average values of the G′ and G″ moduli on time of silicate–polymer systems. (**a**) 20 wt.% of ANa aq. (**b**) 15 wt.% of ANa aq.

**Figure 4 materials-16-07422-f004:**
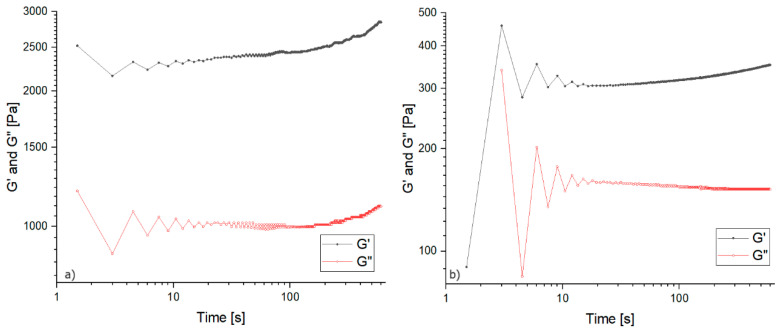
Exemplary graphs obtained during the oscillation test: (**a**) 5%/1:9 and (**b**) 5%/9:1 samples (20 wt.% of ANa aq.).

**Figure 5 materials-16-07422-f005:**
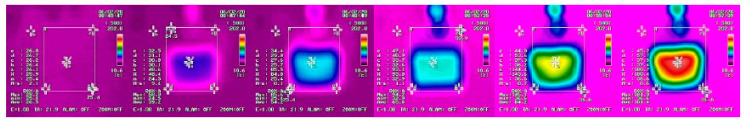
Thermograms of the 20%/1:9 sample (20 wt.% of ANa aq.) obtained during the fire test.

**Figure 6 materials-16-07422-f006:**
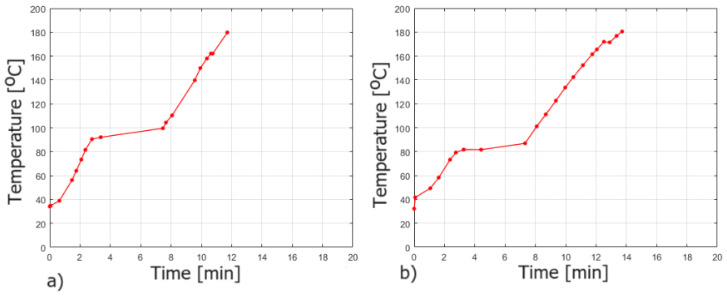
Temperature versus time during the fire test for: (**a**) 5%/1:9 and (**b**) 5%/9:1 (20 wt.% ANa aq.) samples.

**Figure 7 materials-16-07422-f007:**
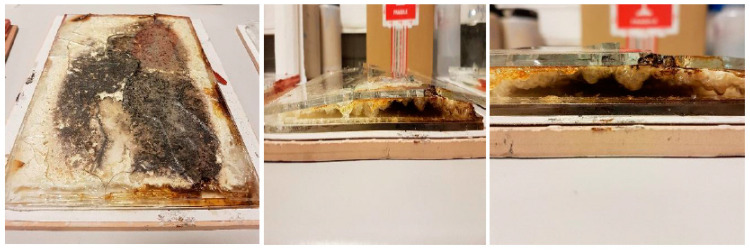
Exemplary glass mold with a hydrogel filling after the fire test (sample 5%/1:9 (20 wt.% of ANa aq.)).

**Figure 8 materials-16-07422-f008:**
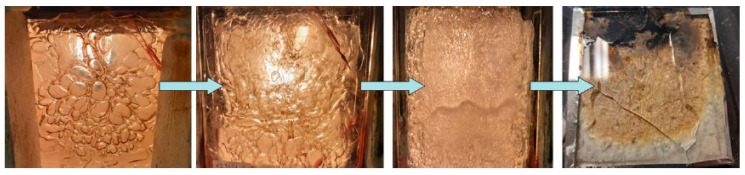
Stages of formation of the fireproof pumice.

**Figure 9 materials-16-07422-f009:**
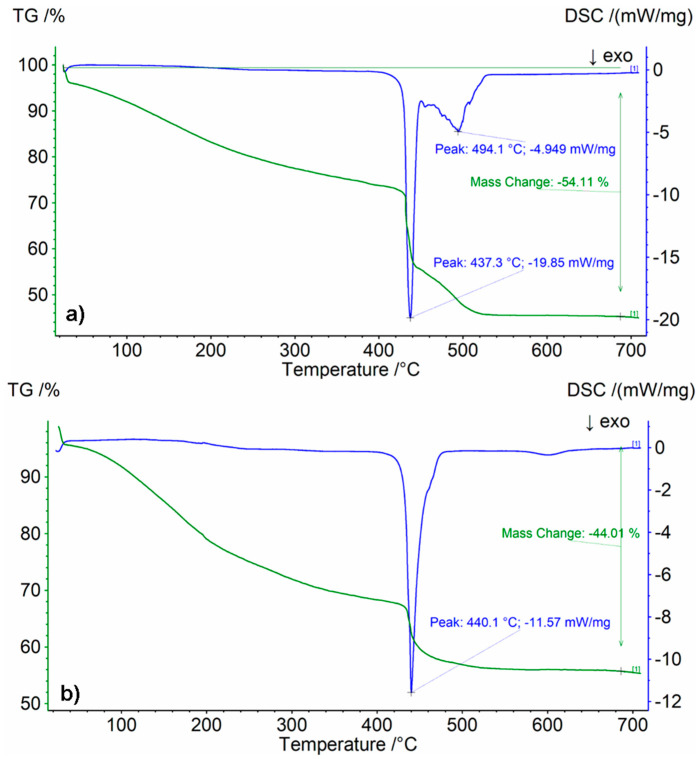
TG/DSC thermograms: (**a**) 5%/1:9 sample (20 wt.% ANa aq.), (**b**) 5%/9:1 sample (15 wt.% ANa aq.).

**Figure 10 materials-16-07422-f010:**
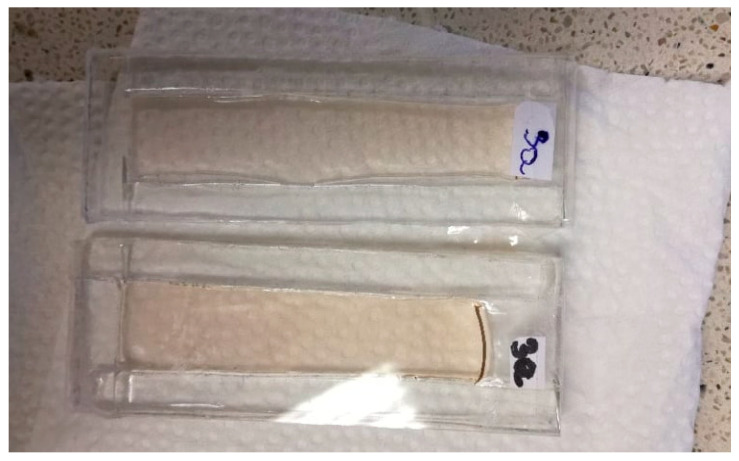
Exemplary hydrogel samples prepared for the translucency measurement (sample 5%/1:9 with 20 wt.% ANa aq.: upper sample—no aging, 2 mm of gel, bottom sample—no aging, 4 mm of gel).

**Figure 11 materials-16-07422-f011:**
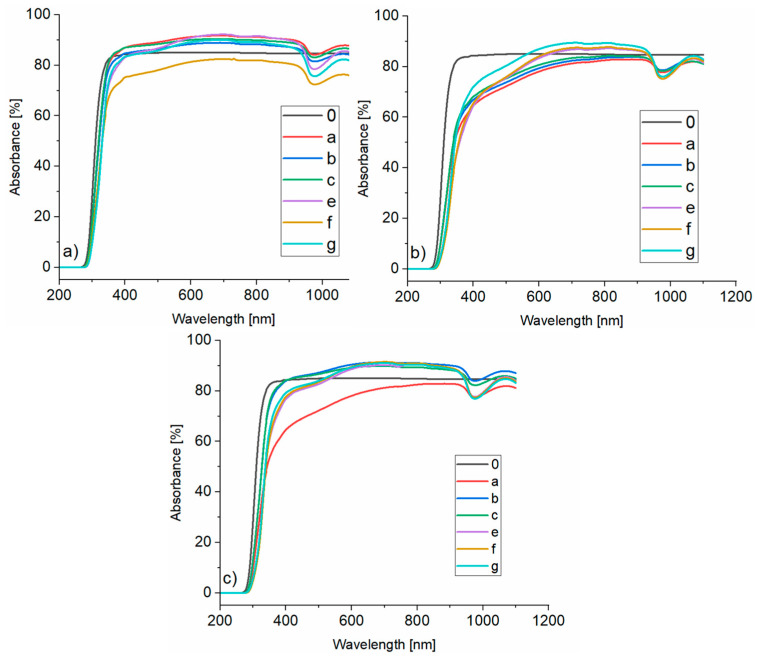
UV/Vis spectra of: (**a**) 5%/1:9 composition with 20 wt.% ANa aq., (**b**) 5%/1:9 composition with 15 wt.% ANa aq., (**c**) 20%/1:1 composition with 15 wt.% ANa aq.

**Figure 12 materials-16-07422-f012:**
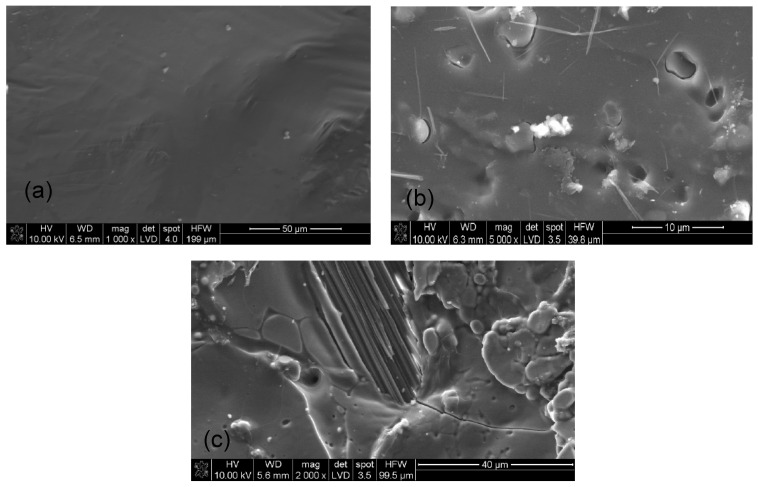
SEM microphotograph of the selected samples before the fire test: (**a**) 5%/1:9 (20%ANa aq.), (**b**) 5%/1:1 (20%ANa aq.), (**c**) 5%/9:1 (20%ANa aq.).

**Figure 13 materials-16-07422-f013:**
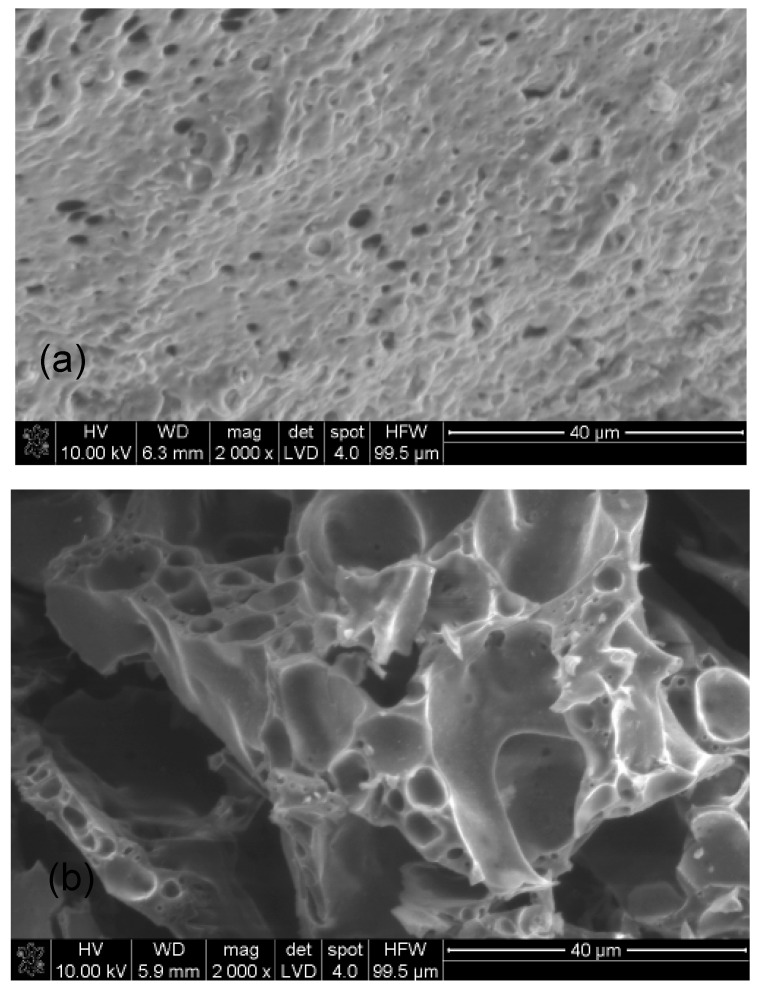
SEM microphotograph of the selected samples after the fire test: (**a**) 5%/1:1 (20%ANa aq.), (**b**) 5%/9:1 (20%ANa aq.).

**Figure 14 materials-16-07422-f014:**
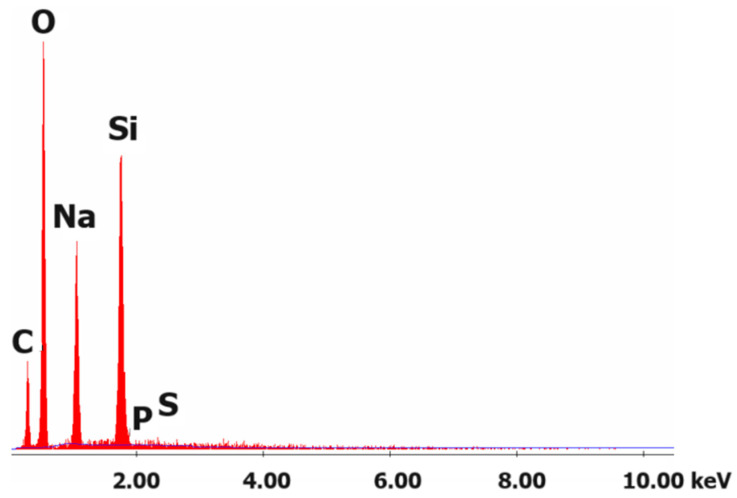
Exemplary EDS analysis of the sample after the fire test (sample 5%/9:1 (20%ANa aq.)).

**Table 1 materials-16-07422-t001:** Compositions of the silicate–polymer hydrogels.

Sample Symbol *	R-100 [wt.%]	WG [wt.%]	ANa aq.** [wt.%]	NNMBA [wt.%]	NTS [wt.%]	KPS [wt.%]
5%/1:9	1	9	90	0.08	0.04	0.04
20%/1:9	2	8	90			
5%/1:1	2.5	47.5	50			
20%/1:1	10	40	50			
5%/9:1	4.5	85.5	10			

* % symbol means the wt.% of the polymer additive. ** 15 wt.% and 20 wt.% concentrations were used as the ANa aq. solutions.

**Table 2 materials-16-07422-t002:** Oscillation parameters of the silicate–polymer hydrogels.

Sample Symbol	G′ [Pa]	G″ [Pa]	φ [°]
20 wt.% sodium acrylate solution			
5%/1:9	9300	3322	19.66
20%/1:9	13,582	4979	20.13
5%/1:1	14,103.5	3888	15.41
20%/1:1	2631.8	1047.2	21.7
5%/9:1	2800.8	1245.8	23.98
15 wt.% sodium acrylate solution			
5%/1:9	1617	533	18.24
20%/1:9	1757.7	469	14.94
5%/1:1	321.7	157.8	26.13
20%/1:1	1522.7	482	17.56
5%/9:1	335	152.75	24.51

**Table 3 materials-16-07422-t003:** Fire insulation times obtained during the fire tests of the silicate–polymer hydrogels.

Type of Sodium Acrylate Solution	Sample Symbol	(I) (180 °C) [min]
20 wt.%	5%/1:9	11:43
	20%/1:9	11:22
	5%/1:1	19:11
	20%/1:1	15:19
	5%/9:1	13:43
15 wt.%	5%/1:9	12:44
	20%/1:9	12:04
	5%/1:1	15:25
	20%/1:1	15:03
	5%/9:1	13:06

**Table 4 materials-16-07422-t004:** Results of TG/DSC analysis of the silicate–polymer hydrogels.

Type of Sodium Acrylate Solution	Sample Symbol	Mass Loss [wt.%]	T_max_ [°C]
20 wt.%	5%/1:9	54.11	437
	20%/1:9	53.98	440
	5%/1:1	48.04	438
	20%/1:1	46.86	435
	5%/9:1	46.70	444
15 wt.%	5%/1:9	54.26	438
	20%/1:9	52.70	437
	5%/1:1	42.50	436
	20%/1:1	47.18	436
	5%/9:1	44.01	440

## Data Availability

Data are contained within the article and Appendix A.

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
