# Peer review of "Effect of Composition on the Physicochemical Properties of Cross-Linked Poly(sodium acrylate)/Sodium Silicate Hydrogels"

_materials, 2023, doi:10.3390/ma16237422_

Round 1
Reviewer 1 Report
Comments and Suggestions for Authors
In this manuscript, the authors prepared the hybrid hydrogels containing the poly(sodium acrylate) network and sodium silicate for a component of a fire-resistant glass. The effect of compositions on the rheological properties, transmittance, and fire-resistant properties was evaluated. The reviewer found one major issue and some minor issues as shown below. After modification of these problems, this manuscript is worth publishing in this journal.
1.
Figure 2: The reviewer cannot understand why G’’ of the samples is initially very low. Sodium water glass is known as a viscous material. Thus, the solution should initially have a large viscosity. The reviewer suspects that there is a problem with the measurement system.
2.
Table 1: “ANa” in the table should be replaced by “ANa aq.” or like that. It is because, for example, the sample coded 5%/1:9 does not contain 90wt% of “sodium acrylate” but contains 90wt% of the sodium acrylate 20wt% (or 15wt%) aqueous solution.
Regarding this, the authors often mention the mass ratio of the water glass and “sodium acrylate” (ex. line 225-226). This “sodium acrylate" should also be replaced by “sodium acrylate solution” or like that.
3.
In line 281, the authors wrote that the sample starts to melt at 70-75C. The reviewer thinks that the sample does not melt because it contains a polymer network. A linear polymer can melt, but a network polymer cannot melt.
4.
Figures 10-12: “Wave number” should be replaced by “Wavelength”.
5.
Line 264-265: “liquid content“ should be replaced by “viscous component”.
Author Response
Joanna Mastalska-Popławska
AGH University of Science and Technology
Faculty of Materials Science and Ceramics
Mickiewicza 30 Av.
30-059 Krakow PL
12th November 2023
Dear Sir/Madam,
At the beginning of my letter I would like to thank you for all of the comments and suggestions. Our article has been corrected in accordance with these remarks. Below is the list of all the changes we made in our revised article (marked in yellow):
- Figure 2: The reviewer cannot understand why G’’ of the samples is initially very low. Sodium water glass is known as a viscous material. Thus, the solution should initially have a large viscosity. The reviewer suspects that there is a problem with the measurement system. - The initial viscosity of the tested mixtures ranged from 10 (for samples with a higher acrylate content) to approximately 100 mPa·s (for samples with a higher water glass content). However, the viscosity at room temperature of the sodium water glass we used was 250 mPa·s. The initial low G" value could also be influenced by the fact that the polymerization reaction was fast and compact gel molecules were formed in the system from the very first seconds.
- Table 1: “ANa” in the table should be replaced by “ANa aq.” or like that. It is because, for example, the sample coded 5%/1:9 does not contain 90wt% of “sodium acrylate” but contains 90wt% of the sodium acrylate 20wt% (or 15wt%) aqueous solution. Regarding this, the authors often mention the mass ratio of the water glass and “sodium acrylate” (ex. line 225-226). This “sodium acrylate" should also be replaced by “sodium acrylate solution” or like that.- the indicated error has been corrected. aq. was added to the abbreviation ANa, and 'solution' was added in the text and in the descriptions of Tables and Figures where required. For example- Table 1, Figure 2.
- In line 281, the authors wrote that the sample starts to melt at 70-75C. The reviewer thinks that the sample does not melt because it contains a polymer network. A linear polymer can melt, but a network polymer cannot melt.- yes, we agree with that. The word ‘melt’ has been changed to ‘soften’.
- Figures 10-12: “Wave number” should be replaced by “Wavelength”.- Figures 10-12 have been changed according to the suggestion; (now they are Figure 11).
- Line 264-265: “liquid content“ should be replaced by “viscous component”.- we have changed that.
Yours faithfully,
Joanna Mastalska-Popławska
Reviewer 2 Report
Comments and Suggestions for Authors This work presents very basic data, poorly orgnanised and shown and it does not have enough novelty. The conditions (i.e. compositions) are not addequatelly selected. In addition, some results and information are missed and it is complicated to follow the flow of the discussion.- The introduction names many studies without giving enough references.
- Line 60, these examples should be in a table with the appropiate references.
- Table 1 is not clear. What do 5 and 20% refer to in the name of the samples? It does not mention that 15 and 20 % of NaA solutions are used. In addition, the selection of the composition is very ramdon and clear tendencies can not be detected and several variables are changed at the same time in every experiment.
- All the graphs of the experiments should be included, at least at the supporting information.
- Table 2 should be ommited and the results should be graphically shown.
- Oscilation measurements are not clearly described in the experimental section.
- Composition C in the UV experiments do not appear in table 1.
- SEM discussion is very poor and not all the samples are shown.
In general, all the sections of the manuscript should be improved.
Comments on the Quality of English Language
English is fine
Author Response
Joanna Mastalska-Popławska
AGH University of Science and Technology
Faculty of Materials Science and Ceramics
Mickiewicza 30 Av.
30-059 Krakow PL
16th November 2023
Dear Sir/Madam,
At the beginning of my letter I would like to thank you for all of the comments and suggestions. Our article has been corrected in accordance with these remarks. Below is the list of all the changes we made in our revised article (marked in green):
- The introduction names many studies without giving enough references.- we have added the omitted references, i.e. 6, 8-10.
- Line 60, these examples should be in a table with the appropiate references.- we have added references to the existing text. Since this is not a review article, we didn’t use the table form.
- Table 1 is not clear. What do 5 and 20% refer to in the name of the samples? It does not mention that 15 and 20 % of NaA solutions are used. In addition, the selection of the composition is very random and clear tendencies can not be detected and several variables are changed at the same time in every experiment.- we have added explanation to the Table 1: *% symbol means the wt.% of the polymer additive; **15 wt.% and 20 wt.% concentrations were used as the ANa aq. solutions. To this article, we have chosen the best samples from the experiment we have prepared, i.e. those which were elastic and have good transparency after the polymerization process. To improve the clarity of the article, we have changed the Figures to make them more comprehensive and legible.
- All the graphs of the experiments should be included, at least at the supporting information. – remaining results from the rheological measurements, fire tests and TG/DSC thermal analysis have been added to the appendix.
- Table 2 should be ommited and the results should be graphically shown.- Table 2 was removed and results are in the form of the graphs (Figure 3).
Figure 3. Dependence of the average values of the G' and G" moduli on time of silicate-polymer systems.
- Oscillation measurements are not clearly described in the experimental section.- we have changed that:” Tests of the oscillation properties were also carried out in the PP-25 system, on the gelled samples in the shape of thin discs with a diameter of about 3 cm and a thickness of 2 mm, prepared 24 hours before the test (samples were cast in the round plastic molds with a diameter of 30 mm). The height of the measurement gap was set at 2 mm, and the entire test was conducted for 10 minutes at ambient temperature, with a constant amplitude of 0.3% and a frequency of 1 Hz. During the measurement, it was observed whether the values of the G' and G'' moduli are changing. Based…”.
- Composition C in the UV experiments do not appear in table 1.- it was a mistake and it should be 5% instead of 10 %: composition 5%/1:9 with 15 wt.% aqueous solution of sodium acrylate.
- SEM discussion is very poor and not all the samples are shown.- We have not shown all the micrographs due to the fact that they are very similar to each other. We have selected three to represent each of the differences discussed in the text. The interpretation of SEM results has been improved: “Based on the photos of the hydrogel samples after drying, it was difficult to determine the relationship between the amount of water glass or polymer. It was only noticed that the more polymer there was in the sample, the smoother the sample surface was. However, when there was more water glass in the sample, the presence of lighter, oval-shaped fields was found (Figure 13 c), what may be the precipitated silica. This may suggest that gelation of the tested systems occurs not only by polymerization, but also by the precipitation of the silicate forms what is influenced by the changing pH. However, no open porosity was found in the examined samples, therefore SEM micrographs of the hydrogels after the fire test were also taken (Figure 14).Some forms of the porosity can be seen in the photos, what may suggest that the tested hydrogels are characterized by the closed porosity. During the contact with a fire, after the water evaporates, the resulting pumice undergoes transformation. Pores enlarge, merge, and then may be damaged under the influence of external factors, such as pressure. Due to the fact that sodium polyacrylate, which is the scaffold of the obtained hybrid hydrogel, has an amorphous structure, the resulting swollen layer has irregular morphology [40-44].”
Yours faithfully,
Joanna Mastalska-Popławska
Reviewer 3 Report
Comments and Suggestions for Authors
1. These elements [sodium silicate and poly(sodium acrylate): is there any alternative materials available] added by the authors in the material design process and their respective advantages should be explained. In this way, the reader will have a clearer idea of the features of this article.
The Figures in the article should be arranged more closely, preferably without blank sections to save space (such as Figures 9 and 10).
The morphology of the hydrogel in the Figure 14 shows some collapse and the pore size is not very uniform. How to prepare the SEM samples? For example, before dry, whether these samples are in a state of swelling equilibrium? Details should be described.
4. SEM descriptions can be strengthened by citing 10.1016/j.cej.2023.141852; 10.1002/adma.202306632.
5. The Abstract section should be strengthened: the important results and main conclusions drawn in this paper should be highlighted and presented in more precise language.
Author Response
Joanna Mastalska-Popławska
AGH University of Science and Technology
Faculty of Materials Science and Ceramics
Mickiewicza 30 Av.
30-059 Krakow PL
16th November 2023
Dear Sir/Madam,
At the beginning of my letter I would like to thank you for all of the comments and suggestions. Our article has been corrected in accordance with these remarks. Below is the list of all the changes we made in our revised article (marked in blue):
- These elements [sodium silicate and poly(sodium acrylate):is there any alternative materials available] added by the authors in the material design process and their respective advantages should be explained. In this way, the reader will have a clearer idea of the features of this article.- appropriate explanation have been added to the last paragraph of the introduction: ‘As a polymer component, from among the available vinyl monomers such as acrylic monomers, methacrylates, vinyl acetates and vinyl alcohol, we chose sodium acrylate due to the fact that its polymer can bind significant amounts of water and polymerize in the presence of other chemical substances, and the polymerization process itself it's not complicated. In the case of the silicate component, we chose sodium water glass because it is non-toxic, cheap and easily available, and during cross-linking of vinyl polymers, silicon gel precipitates from it and a mineral-organic hydrogel with increased durability and elasticity is created [26].’.
- The Figures in the article should be arranged more closely, preferably without blank sections to save space (such as Figures 9 and 10). – we have rearranged Figures with UV-Vis spectra and now they are in one Figure, i.e. Figure 11.
- The morphology of the hydrogel in the Figure 14 shows some collapse and the pore size is not very uniform. How to prepare the SEM samples? For example, before dry, whether these samples are in a state of swelling equilibrium? Details should be described.- details were added to the description of the SEM method: ‘Hydrogel samples before the fire tests were dried at room temperature, frozen using liquid nitrogen and immediately crushed in a metal mortar. They were then dried again at room temperature and tested. Pumice, which was used as samples after the fire tests, was collected from between the glass panes and then tested.’
- SEM descriptions can be strengthened by citing 10.1016/j.cej.2023.141852; 10.1002/adma.202306632.- we have strengthened the SEM analysis by adding the better description: ‘During the contact with a fire, after the water evaporates, the resulting pumice undergoes transformation. Pores enlarge, merge, and then may be damaged under the influence of external factors, such as pressure. Due to the fact that sodium polyacrylate, which is the scaffold of the obtained hybrid hydrogel, has an amorphous structure, the resulting swollen layer has irregular morphology [40-44].’ (subchapter 3.5, second paragraph).
- The Abstract section should be strengthened: the important results and main conclusions drawn in this paper should be highlighted and presented in more precise language.- the abstract section has been rewritted: ‘The paper presents the results of research on the influence of the components of the composition on the physicochemical properties of cross-linked poly(sodium acrylate)/sodium silicate hydrogels. The results of the rheological measurements showed that an increasing share of polyacrylate in the sample causes a shift of the cross-over point towards longer times and improves mechanical properties. In turn, increasing the share of sodium silicate has a positive effect on extending the fire insulation time (I). The TG/DSC analysis indicated that if the sample contained more than 50 wt.% of polymer, the thermal decomposition was a three-stage process, while if the sample contained more water glass, the decomposition was two-stage. UV-Vis measurements and SEM analysis confirmed, inter alia, that increasing the share of polyacrylate causes improvement of the optical parameters and homogeneity of the tested hydrogels. The presented results suggest that by controlling the share of the main components of the silicate-polymer hydrogels, it is possible to control their mechanical, optical and thermal properties. Multiplication of the layers or increasing their thickness improves their fire-retardant properties.
Yours faithfully,
Joanna Mastalska-Popławska
Reviewer 4 Report
Comments and Suggestions for Authors
The paper entitled “Effect of the composition on physicochemical properties of cross-linked poly(sodium acrylate)/sodium silicate hydrogels” deals with an interesting research field, that of developing efficient construction materials. The results presented within this paper are valuable, however some minor aspects should be considered before publication:
- Figures 2, 3,4 ,5 present example graphs for the analyzed samples, thus for consistency authors should consider the same samples to show.
- Also, a short explanation about viscoelastic behavior given by the phase shift angle should be introduced.
- Figure 6 caption should contain the sample name.
- Regarding the DSC results, authors state:” Silicates undergo only polymorphic transformations at the tested temperatures” but there is no data about this transition nor in table 5 or in the curves from figure 8. Also they say “No glass transition of the polymer was observed for any of the analyzed samples” , I recommend testing the polymer and the water glass separately, it could give a clear insight about explaining these thermal processes, which are highly important in this application.
Comments on the Quality of English LanguageThere are some minor typo issues (e.g. in the abstract polycarylate).
In some cases rephrasing should be considered : e.g. page 9 rows 279-299
Author Response
Joanna Mastalska-Popławska
AGH University of Science and Technology
Faculty of Materials Science and Ceramics
Mickiewicza 30 Av.
30-059 Krakow PL
16th November 2023
Dear Sir/Madam,
At the beginning of my letter I would like to thank you for all of the comments and suggestions. Our article has been corrected in accordance with these remarks. Below is the list of all the changes we made in our revised article (marked in pink):
- Figures 2, 3,4 ,5 present example graphs for the analyzed samples, thus for consistency authors should consider the same samples to show.- according to this, we have changed Figures 4 and 6, so all the mentioned Figures show results for 5%/1:9 and 5%/9:1 samples.
- Also, a short explanation about viscoelastic behavior given by the phase shift angle should be introduced.- explanation has been added to the subchapter 3.1.2, second paragraph: ‘However, no trend was found depending on the composition of the samples, i.e. comparing the share of total polymer and water glass content in terms of the phase shift angle values. The obtained hydrogels are not liquid, and due to the presence of a three-dimensional polyacrylate network, even in the minimum amount for the 5%/9:1 sample, they are highly flexible materials. If this mixture were stripped of the cross-linking agent, a highly viscous liquid would be obtained, with little evidence of gel structure from the precipitated silicate gel.’.
- Figure 6 caption should contain the sample name.- we have added the sample name: sample 5%/1:9 (20 wt.% of ANa aq.).
- Regarding the DSC results, authors state:” Silicates undergo only polymorphic transformations at the tested temperatures” but there is no data about this transition nor in table 5 or in the curves from figure 8. Also they say “No glass transition of the polymer was observed for any of the analyzed samples” , I recommend testing the polymer and the water glass separately, it could give a clear insight about explaining these thermal processes, which are highly important in this application.- We have removed first state from the article and we have added explanation to the second sentence, i.e. “No glass …”.: “This was caused by the overlap of thermal effects resulting from the transformation of both hydrogel components.”. We have tested the polymer and water glass separately before and the results are described in the following article (Reference 21 in the revised article): Mastalska-Popławska, J.; Stempkowska, A.; Habina-Skrzyniarz I.; Krzyżak, A.T.; Rutkowski, P.; Izak, P.; Rudny, J.; Gawenda, T. Water interactions in hybrid polyacrylate-silicate hydrogel systems. Materials 2020, 13.
- There are some minor typo issues (e.g. in the abstract polycarylate).- we have corrected the mentioned typos.
- In some cases rephrasing should be considered : e.g. page 9 rows 279-299- we have rephrased the mentioned paragraph: Analyzing the influence of the composition on the fire insulation results, it was noticed that the more water glass was in the sample, the longer the fire insulation time was. This can be explained by the fact that water glass is a fire retardant, which, by losing water and transforming into a swollen pumice upon the contact with high temperatures, extends the insulation time. However, the best results were obtained for the samples where the content of the water glass and sodium polyacrylate was similar, what allows to assume that the structure of the resulting hybrid hydrogel may also have an impact on the fire insulation and in this case it was the most durable (mechanical strength has been confirmed by rheological tests). The thickness of the gel layer may also have an impact on the fire insulation time. The thicker it is, the longer this time will be [15, 35, 36].
It was noticed, when analyzing compositions with 15 wt.% and 20 wt.% of aqueous solution of sodium acrylate, that both gel systems increase significantly in volume (up to 5 times)during the measurement (Figure 7). They start to soften and "boil" at a temperature of about 70-75 °C. The more water was in the gel, the greater this effect was. However, the gel turns into foam very quickly and at a temperature of about 90 °C solid begins to form and fire-resistant pumice is created (Figure 8). At a temperature of about 80 °C, the so-called plateau was observed, what is confirmed in Figure 6. It can be concluded that in the case of the developed silicate-polymer hydrogels, the gel melting effect cannot be eliminated, but it is very limited.
Yours faithfully,
Joanna Mastalska-Popławska
Round 2
Reviewer 1 Report
Comments and Suggestions for Authors
The authors have modified the problems. For the rheological test, even though the reviewer still does not understand the physical reason for the result, the authors show their ideas for this problem. The reviewer thinks that the modification is enough and the paper is now worth publication in this journal.
Author Response
Joanna Mastalska-Popławska
AGH University of Science and Technology
Faculty of Materials Science and Ceramics
Mickiewicza 30 Av.
30-059 Krakow PL
25th November 2023
Dear Sir/Madam,
I would like to thank you for acceptance the improvement of the article.
Yours faithfully,
Joanna Mastalska-Popławska
Reviewer 2 Report
Comments and Suggestions for Authors
The authors have addressed the most relevant suggestions so the paper can be accepted in its present form.
Author Response

(The authors gave the same response as above.)
